# Oncolyic Virotherapy for Prostate Cancer: Lighting a Fire in Winter

**DOI:** 10.3390/ijms232012647

**Published:** 2022-10-21

**Authors:** Gongwei Wang, Ying Liu, Shuoru Liu, Yuan Lin, Cheng Hu

**Affiliations:** 1Department of Urology, The Third Affiliated Hospital of Sun Yat-sen University, Guangzhou 510630, China; 2Department of Infectious Diseases, The Third Affiliated Hospital of Sun Yat-sen University, Guangzhou 510630, China; 3Department of Pharmacology, Sun Yat-sen University, Guangzhou 528478, China

**Keywords:** oncolytic vrius, prostate cancer, immunotherapy, combined medication

## Abstract

As the most common cancer of the genitourinary system, prostate cancer (PCa) is a global men′s health problem whose treatments are an urgent research issue. Treatment options for PCa include active surveillance (AS), surgery, endocrine therapy, chemotherapy, radiation therapy, immunotherapy, etc. However, as the cancer progresses, the effectiveness of treatment options gradually decreases, especially in metastatic castration-resistant prostate cancer (mCRPC), for which there are fewer therapeutic options and which have a shorter survival period and worse prognosis. For this reason, oncolytic viral therapy (PV), with its exceptional properties of selective tumor killing, relatively good safety in humans, and potential for transgenic delivery, has attracted increasing attention as a new form of anti-tumor strategy for PCa. There is growing evidence that OV not only kills tumor cells directly by lysis but can also activate anticancer immunity by acting on the tumor microenvironment (TME), thereby preventing tumor growth. In fact, evidence of the efficacy of this strategy has been observed since the late 19th century. However, subsequently, interest waned. The renewed interest in this therapy was due to advances in biotechnological methods and innovations at the end of the 20th century, which was also the beginning of PCa therapy with OV. Moreover, in combination with chemotherapy, radiotherapy, gene therapy or immunotherapy, OV viruses can have a wide range of applications and can provide an effective therapeutic result in the treatment of PCa.

## 1. History of Oncolytic Virotherapy

Oncolytic virotherapy uses replication-competent viruses that are able to lead to lysis of tumor cells after infecting and replicating. Viruses with a lytic effect have been identified, including adenovirus (Ads), herpes simplex virus (HSV), cowpox virus, vesicular stomatitis virus (VSV), respiratory intestinal orphan virus (EWV), Newcastle disease virus (NDV), coxsackievirus, measles virus (MeV), Sendai virus (Hemagglutinating virus of Japan), etc. (Table 1). Parallel to the emergence of virology, which began in the late 19th century, oncolytic virotherapy has an ancient history [1]. The remission of a 42-year-old woman with leukemia after contracting influenza in 1896 and the improvement in a patient with advanced cervical cancer after rabies vaccination in 1912 further demonstrate the connection between viruses and tumor regression [2,3]. Another case is transient regression of lymphocytic leukemia after varicella infection in a 4-year-old boy [4].

Inspired by these anecdotal observations, scientists have gradually explored oncolytic virotherapy in a number of clinical trials in the middle of the past century. The hepatitis B virus was the first to be tested in Hodgkin disease patients in 1949, and results showed that anti-cancer effects were observed in 7 out of 13 patients for ≥1 month [5]. In the 1970s, treatment with wild-type mumps virus showed amazing effectiveness, such that 39 of 90 patients with various advanced cancers showed complete tumor regression or >50% reduction in tumor volume [1]. Although oncolytic virotherapy is a potentially exciting new anti-tumor treatment, two major challenges remain in ensuring that oncolytic viruses are safe enough and that enough viruses are delivered to tumor tissue to ensure anti-tumor efficacy. Local injection of the virus ensures a sufficient amount of virus in the tumor tissue but does not apply to certain cancers, such as glioma and pancreatic cancer. As with other chemotherapeutic agents, intravenous administration is the ideal method of treatment for oncolytic virotherapy, ensuring that the virus reaches cancerous sites anywhere in the body but may cause a systemic and intense antiviral response and does not guarantee that enough virus will accumulate in the tumor tissue. Owing to emerging chemotherapy alternatives, virotherapy trials declined in the 1970s until the 1990s, when advances in molecular virology gave renewed impetus to virotherapy. With the genetic engineering modification, attenuated virus mutants with oncolytic activity restricted lytic replication to tumor cells only, which reduces virus-induced toxicity in normal tissues and, accordingly, remains one of the main strategies in the development of safe virotherapy.

## 2. General Anti-Tumor Molecular Mechanisms of OVs

### 2.1. OVs Lyse Tumors Directly but Not Ordinary Cells

Possessing the ability to replicate through the lytic cycle, OVs can directly destroy the tumor cells of the host and infect neighboring cells with newly formed virions. Not only associated with cellular antiviral response components (PKR, Toll-like receptor TLR, retinoic acid-inducible gene 1 RIG-1, IFN, etc.), this capacity also depends on virus type, dose, natural and induced virulence, and susceptibility of cancer cells to different forms of cell death (apoptosis, necrosis, cytokinesis and autophagy). On the contrary, there are multiple signaling pathways that act to detect and clear viral particles for normal cells. Defects in these pathways in tumor cells are key to their inability to clear the virus and be lysed by the virus (Figure 1) [6].

When OV infects normal cells, the viral components trigger antiviral immune response through multiple mechanisms that involve stimulation of the intracellular Toll-like receptors (TLRs), which induce secretion of interferon type I (IFN-I), which triggers antiviral immune response. Expressed in a variety of cells, TLRs are pattern recognition receptors on the cell surface and in cells that could be activated by pathogen-associated molecular patterns (PAMPs), with repetitive sequences common to pathogenic bacteria and viruses. These PAMPs include viral capsids, DNA, RNA and viral proteins. TLR will then activate TNF-related factor 3 (TRAF3). RIG-1 recognizes viral nucleic acids and is activated, which together with TRAF3 continues to activate downstream factors, such as IFN-related factor 3 (IRF3) and IRF7, thereby further activating the JAK-STAT (Janus kinase signal transducer and activator of transcription) pathway, which coordinates the antiviral machinery in infected cells. Factors downstream of this pathway, such as IRF7, enhance local IFN release and promote IFN-mediated antiviral responses. The TLR pathway and IFN pathway are interconnected to trigger the antiviral immune response in normal cells (Figure 1) [7,8,9]. In addition, TLR signaling activates dendritic cells (DCs), as well as macrophages Mø to secrete IL-12—a cytokine that induces the conversion of helper T cells to the Th1 phenotype [10], which plays a role in the activation of anti-tumor immunity by OVs.

By binding to the IFN receptor (IFNR), IFN activates PKR, an intracellular protein kinase that recognizes double-stranded RNA and other viral components. When PKR is activated, it terminates cellular protein synthesis and promotes rapid cell death and virus clearance [11,12]. Moreover, IFN signaling through the IFNAR receptor ultimately leads to the expression of interferon-stimulated genes (ISG). When viral proteins are modified by ISG, their own localization and protease activity are affected, oligomerization and geometry are disrupted, and interactions with host proteins or other viral proteins are destroyed, which eventually leads to the reduction of virus replication.

Immune deficiency is often present in tumor cells, meaning that abnormalities in the IFN pathway and PKR activity can interfere with viral clearance, with rapid viral replication and lysis of host cells, resulting in specific killing of tumor cells by OVs. OVs can manipulate various signaling pathways within tumor cells to prevent apoptosis, giving the viruses more time to complete their life cycle. OVs ultimately induce cell death, and both the form of cell death and the release of danger signals from virally infected cells can greatly assist in the induction of a host immune response, which not only directly clears tumor cells but also sets the stage for the initiation of a systemic immune response.

### 2.2. OVs Induce Anti-Tumor Immunity

#### 2.2.1. OVs Induce Local and Systemic Anti-Tumor Immunity

Possessing a broad ability to activate the body′s immune system, OVs can cause immunogenic cell death (ICD) of tumor cells (Figure 2) by lysing cells, releasing soluble antigens, danger signals and IFN thereby stimulating anti-tumor immunity [13]. OVs also induce T cell responses in tumors, which can promote local inflammation and recruit CD4^+^ and CD8^+^ T cells associated with an anti-tumor response [14,15,16]. In the meantime, the “neoantigens” that are transiently expressed during normal tissue repair after local inflammation induce immune surveillance-related immune responses, which also help clear tumor cells that express “neoantigens” but are not infected with the virus. After tumor cells are lysed, the local release of cytotoxic perforin and granzyme may kill nearby tumor cells, whether they are infected by viruses or not. This phenomenon of local adjacent tumor cell death through immune inflammation and other means is called the bystander killing effect. This induction of local, systemic innate, and tumor-specific immune responses appears to be a key factor in tumor eradication by OVs, as opposed to the direct lysis of tumor cells.

When cells infected with OVs die, they release tumor-associated antigens (TAAs), viral PAMPS, and cellular damage-associated molecular patterns DAMPs (e.g., heat shock proteins, high mobility group B1 (HMGB1), calreticulin (CALR), ATP and uric acid, etc.) and cytokines (such as type I IFN, TNF-α, IFN-γ and IL-12). Among these, CALR, ATP and HMGB1 are considered to play a key role among all three DAMPs.

##### CALR

CALR is thought to be transferred to the cell surface early in the ICD process (Figure 2) [17,18,19,20,21], and upon binding to low-density lipoprotein receptor-related protein 1 (LRP1, also known as CD91), it transmits phagocytic signals to APCs, such as dendritic cells (DCs), consequently increasing the ability to phagocytize dead cells [22,23,24,25,26,27,28,29,30,31]. The endoplasmic reticulum stress response is the basis of CALR exposure [32]. Interestingly, phagocytosis stimulation by CALR is neutralized by CD47 expression in a large number of solid and hematopoietic tumors [24]. This indicates that tumor cells can avoid the effect of CALR through CD47, so as to prevent immune killing caused by physiologically dead tumor cells via CALR in the process of tumorigenesis.

##### ATP

ATP released extracellularly by tumor cells following the direct lysis of OVs is a powerful chemoattractant (Figure 2). It promotes not only the recruitment of immune cells to ICD sites but also the differentiation of these immune cells. These functions are associated with the G-protein coupled purinergic receptor 2 (receptor P2Y, G-protein coupled, 2; P2RY2) [33,34,35,36]. In addition, extracellular ATP promotes the activation of the NLR family and pyrin-containing structural domain 3 (NLRP3) inflammasome in APCs, thereby stimulating the release of IL-1β and IL-18, which are two significant pro-inflammatory cytokines involved in the immune response [37,38,39,40,41,42,43,44,45,46,47]. This immune process can interfere if pericellular ATP is converted to ADP or AMP using recombinant adenosine triphosphate bisphosphatase (apyrase, an ATP degrading enzyme) or exonucleoside triphosphate diphosphate hydrolase 1 (ENTPD1, i.e., CD39) [48]. Similar to the above anti-CALR effect, tumor cells overexpress CD39 and the 5′-nucleotidase ecto (NT5E, i.e., CD73) to convert ATP to adenosine and thus exert immunosuppressive effects to ensure their own growth [49,50,51,52,53,54].

##### HMGB1 

When HMGB1 is released extracellularly (Figure 2), it binds to several receptors (TLR2, TLR4, AGER) on the surface of immune cells, mediates strong pro-inflammatory effects [55,56,57,58,59,60,61,62,63], and can also exert immune chemotactic activity by forming complexes with chemokine (CXC motif) ligand 12 (CXCL12) [64]. Also, endogenous HMGB1 promotes autophagy by interfering with the reciprocal inhibition between the central sub-temporal regulator beclin 1 (BECN1) and the anti-apoptotic protein B-cell CLL/lymphoma 2 (BCL2) [65,66,67], so we speculate that HMGB1 release may contribute to the ICD inducer-induced cellular autophagic response.

##### IL-12 

In addition to the above three substances (CALR, HMGB1, ATP), interleukin-12 (IL-12), a potent anti-cancer cytokine, is also important in the induction of anti-tumor immunity by OVs. IL-12 is normally produced by phagocytes (monocytes/macrophages, neutrophils) and dendritic cells in response to pathogens and directly activates innate immune cells (NK cells, NK-T cells) and adaptive immune cells (CD4^+^ cells, CD8^+^ cells). IL-12 initiates T cells and enhances their survival, promotes Th1 differentiation and enhances T cell, NK cell and NK-T cell effector functions. As mentioned earlier, tumor immunotherapy requires the involvement of T cells, while OVs and IL-12 can act synergistically to increase the recruitment of CTL cells into the tumor microenvironment (TME). IL-12 also induces the secretion of IFN-γ, which acts directly on tumor cells in the tumor microenvironment as well as on stromal and endothelial cells, and its signaling leads to (1) increased MHC I processing and presentation (enhancing tumor recognition by T cells) and (2) induction of chemokines IP-10 and MIG, recruiting innate and adaptive immune effectors, leading to (i) altered extracellular matrix remodeling, including inhibition of matrix metalloproteinase expression, which reduces angiogenesis and tumor invasion, and (ii) reduced expression of adhesion molecules in endothelial cells, which may further limit angiogenesis [68,69,70,71].

These substances promote the maturation of antigen-presenting cells’ APCs (e.g., DCs), which promote the differentiation of T cells towards specific CD4^+^ T cells and CD8^+^ T cells [72]. CD8^+^ T cells act as cytotoxic T lymphocytes (CTL) that, when activated, metastasize to the site of tumor growth and recognize tumor-specific antigens to mediate anti-tumor immunity [9]. This process is vital for systemic anti-tumor immunity, and the importance of tumor-specific CD8^+^ T cells in mediating immune tumor elimination of OVs has been demonstrated in many pre-clinical studies [16,73,74,75,76]. The type I IFN and DAMP released by OVs after tumor lysis also mediate the activation of natural killer (NK) cells, further promoting an anti-tumor immune response as part of innate immunity [77]. Moreover, OV infection leads to downregulation of major histocompatibility complex (MHC) class I expression on the host cell surface [78,79], and NK cells can kill target cells with downregulated MHC class I expression, which is common in cancer cells [80,81]. Notably, IFN-γ released from tumor cells by OVs upregulates the expression of MHC class I on the surface of nearby cancer cells, thereby enhancing the CTL-mediated immune response (Figure 3) [82,83,84,85]. 

Overall, TAA, PAMPS, DAMP and cytokines released by OVs from directly lysed cells not only cause ICD in local and distant tumor cells by inducing inflammation and activating adaptive T cells, but also promote the recruitment of APCs to the site of ongoing ICD, triggering adaptive immunity to further enhance ICD and creating positive feedback [86]. These actions lead to changes in the composition of the TME, accordingly disrupting the original tumor cell growth environment and laying the foundation for long-term cancer eradication.

#### 2.2.2. OVs Counteract Tumor Immune Evasion

Tumor immune evasion is considered one of the “hallmarks of cancer” and represents a major direction for new cancer therapy [87]. Cancer cells avoid the destruction of immune-mediated responses through complex mechanisms, including the progressive development of an immunosuppressive environment within the tumor and the selection of tumor variants against immune effectors (sometimes called “immunoediting”) [88]. There are currently six known mechanisms of immune evasion: (1) Fas/Fasl-mediated immune evasion; (2) decreased immunogenicity and antigen modulation of tumor-associated antigens; (3) decreased or absent expression of MHC molecules on tumor cell surfaces; (4) expression of immunosuppressive molecules, such as PD-L1 and CTLA-4 on the surface of tumor cells to disable T cells; (5) release of tumor antigen molecules that bind to antibodies, or the NK cells and macrophages through the FC segment of antibodies to block the ADCC effect; and(6) tumor-mediated autocrine or paracrine production of immunosuppressive cytokines, such as IL-10, transforming growth factor-β (TGF-β) and indoleamine-2,3-Dioxygenase (IDO) to promote the recruitment of immunosuppressive cells, such as tumor-associated macrophages and myeloid-derived suppressor cells (MDSCs), to tumor growth sites [89,90,91,92,93]. Limited circulating immune effector cell infiltration is also characteristic of the immunosuppressive tumor microenvironment. Overall, the mechanism by which OVs counteract tumor immune evasion is via poor T-cell infiltration, low tumor mutational burden (TMB), and the establishment of a non-immunogenic tumor microenvironment [94]. 

Remarkably, based on the spatial distribution of CTLs in the tumor microenvironment (TME), tumors can be classified into three basic immunophenotypes: immune inflammation, immune rejection and immune desert. The immune inflammation phenotype, also known as the immunogenic phenotype, is termed a “hot” tumor and responds well to immunotherapy. The immune rejection and immune desert phenotypes are collectively referred to as non-immunogenic phenotypes, which are also known as “cold” tumors and are generally less responsive to immunotherapy [95,96,97]. Although therapeutic monoclonal antibodies (mAb) against immune checkpoints (ICP), such as anti-PD-1/PD-L1 agents have offered a new approach to tumor therapy recently, immune checkpoint inhibitors (ICI) generally have a limited impact on tumors because of the low level of T cell infiltration within the tumor immune microenvironment (TIME).

OVs restrict the immunosuppressive tumor microenvironment by modifying various mechanisms that alter the cytokine environment and immune cell types [98,99]. As previously described, it has been shown that upregulation of peripheral cellular MHC and costimulatory molecules can be observed in tumor cells following oncolytic virus infection. This upregulation promotes immune-mediated tumor cell recognition and eradication and triggers tumor-associated antigen exposure and epitope spreading, which in turn recruits immune cell populations and transforms ‘cold’ tumors with a ‘non-T-cell inflammatory’ phenotype into ‘hot’ tumors with a ‘T-cell inflammatory phenotype’ [100,101]. Furthermore, the expression of multiple immunostimulatory transgenes within the recombinant OV genome may reshape the tumor immune microenvironment with minimal toxicity through direct access to the tumor niche. One such transgene is the granulocyte-macrophage colony-stimulating factor (GM-CSF), an immunostimulatory molecule that recruits specialized APCs, including DCs, promotes cancer antigen presentation, recruits immune cells to mature, and activates NK cells and tumor antigen-specific T cells. Simultaneously, the combination of the release of “neoantigens” after the tumor cells are killed and the bystander-killing effect will also “heat up” the tumor immune microenvironment and thus better counter tumor immune evasion [94,95,96,97].

OV-induced alteration of the tumor immune microenvironment was observed not only at the site of virus injection but also in non-virally injected lesions, suggesting that the mechanism of countering tumor immune evasion can be systemic and can be the subsequent part of OVs inducing local and anti-tumor immune responses to kill tumor cells. Studies have shown that this alteration enhances the effect of ICIs, such as anti-CTLA-4mAb, anti-PD1/PD-L1mAb [16,102,103,104,105], and that both OVs and ICIs act on the majority of tumors without targeting specific cancer types, suggesting the potential of OVs to be used in combination with tumor immunotherapy.

## 3. Epidemiology and Current Therapy of PCa

### 3.1. Global Epidemiology of PCa

Effective PCa treatment is a hot topic of interest to urologists around the world. Although the new coronavirus (COVID-19) has ranked first in deaths from all types of diseases since its globalization in 2019, cancer is still one of the major public health problems worldwide. Globally, PCa is the second most common solid tumor in men, with approximately 13 million new cases diagnosed worldwide each year. According to American Cancer Society statistics in 2022, PCA ranks first in frequency of occurrence (27% of all new cancer cases) and second in number of deaths (11% of 325 all cancer deaths) among men in the United States, which means that PCa remains the most prevalent tumor in the genitourinary system [106,107]. The incidence of PCa varies considerably between continents and Asia is traditionally considered to be a low incidence region [108]. Nonetheless, due to economic development, increased life expectancy and westernized lifestyles, the incidence of PCa in Asia is rising rapidly [109,110]. With almost no symptoms at first, PCa is difficult to detect in its early stages, and at the time of diagnosis, the cancer may already be metastasized and in a terminal stage. According to statistics, around 10 million men are currently diagnosed with PCa, 700,000 of whom have metastases, and the death rate is expected to more than double by 2040 [111,112,113]. PCa that is in the terminal stage have usually already metastasized to, e.g., bone or present as castration-resistant cancer form (CRPC), often difficult to treat. 

The median overall survival (OS) for patients with mCRPC ranges from 13 to 32 months, with a 5-year survival rate of 15%. Therefore, the development of new treatments for PCa, especially for intermediate and advanced PCa, is essential. 

### 3.2. Overview of the Current State of PCa Therapy

Treatment options for PCa include active surveillance (AS) and watchful waiting, surgery, endocrine therapy, chemotherapy, radiotherapy, immunotherapy, etc. Summarizing the 2020 edition of European Urology and the guidelines of the National Comprehensive Cancer Network (NCCN) [114,115,116], as for the traditional treatment, it is now considered that ① active surveillance (AS) as an assessment tool using PSA, DRE, biopsy and other indicators can be applied to patients with low-risk PCa (tumors confined to the peritoneum) to reduce overtreatment and not compromise opportunities for cure. ②Radiotherapy is indicated for low-risk PCa patients with a prostate volume of <50 mL, or in combination with ADT for patients with intermediate or high-risk PCa. ③ Patients with PCa whose tumors are removable and not invading the urethral sphincter have the opportunity to undergo RP, but for patients with intermediate risk, high risk and locally advanced PCa, postoperative adjuvant therapy is also required. ④ Endocrine therapy (androgen deprivation therapy, ADT) is indicated as adjuvant therapy after RP and in combination with the chemotherapeutic agent doxorubicin for metastatic prostate cancer (mPCa), castration-resistant PCa (CRPC) and mCRPC. It has been shown that sex hormones play an important role in the pathogenesis and progression of PCa and that tumor progression can be better controlled after castration [117]. 

As for the new treatment, ⑤ the radiopharmaceutical radium 223 is mainly used as a palliative treatment for bone pain in patients with bone metastases and although it is effective in relieving pain, it has no impact on survival [118]. ⑥ As for targeted drugs, the PARP inhibitors olaparib and rucaparib have also been approved for the treatment of mCRPC in patients with mutations in DNA homologous recombination repair genes (e.g., BRCA1, BRCA2, ATM, PALB2, FANCA, RAD51D, CHEK2) [119,120,121,122,123,124,125,126,127,128,129,130,131,132,133]. ⑦ For immunotherapy, Sipuleucel-T (a dendritic cell-based autologous vaccine) is thought to increase OS in asymptomatic or mildly symptomatic mCRPC patients. However, due to doubts about its efficacy, manufacturing difficulties and associated costs, this treatment has only been approved by the FDA and not by any European regulatory body, so its use in the clinical setting remains relatively limited [118]. ⑧ With the rise of ICI therapies in recent years, anti-PD-1 antibody (pembrolizumab), which has been classified as a Class 2B recommendation, has been used to treat advanced PCa with high microsatellite instability (MSI-H)/defective mismatch repair (dMMR) [134,135,136,137,138,139,140,141,142,143,144]. 

Current treatment for patients with low-risk PCa is satisfying, but some subpopulations of tumor cells in progressive PCa can alter their nuclear androgen receptor expression levels and function through androgen receptor (AR) increase and hypersensitivity, AR mutations, co-activator/co-inhibitor mutations, androgen non-dependent AR activation and intratumoral androgen production, allowing cancer to survive with low androgen levels or in the absence of androgens (i.e., as CRPC) [145], making conventional therapies, particularly ADT, less effective. At this time, anti-tumor immunotherapy may be a superior alternative. Immunotherapy for tumors aims to enhance natural defenses to eliminate malignant cells or impair their phenotype and function in the long term, representing a major breakthrough in cancer treatment. 

Anti-tumor immunotherapy has long been in the PCa treatment guidelines [114,115,116], but unfortunately, immune checkpoint inhibitors have not yet shown efficacy in PCa, despite altering clinical outcomes in other solid tumors. Ipilimumab, an anti-cytotoxic T-lymphocyte-associated 4 (CTLA4) checkpoint inhibitor, was studied in two phase III clinical trials of mCRPC, both of which showed no improvement in OS in mCRPC patients [146,147]. Pembrolizumab showed high response rates in tumors with mismatch repair defects, leading to FDA approval but did not have PCa specificity. Actually, some studies have shown that only 2–12% of PCa have microsatellite instability and hypermutation status that qualify for anti-PD-1 antibody therapy [148,149]. At the same time, the EAU notes that all patients who receive treatment for mCRPC will eventually progress, which suggests that current therapies are not able to provide a satisfactory survival benefit for mCRPC patients—the median overall survival (OS) for mCRPC patients ranges from 13–32 months, with a 5-year survival rate of 15%. In general, PCa in metastatic form, e.g., with bone metastases in intermediate or advanced stages, or in castration-resistant form (CRPC), is difficult to cure. Therefore, the development of new treatments for advanced PCa, especially more effective anti-tumor immunotherapy, is essential.

## 4. Oncolytic Virotherapy for PCa

### 4.1. Pre-Clinical Study of OVs for PCa

There are a variety of OVs currently used in preclinical studies, with Ads predominating in the early years. In fact, as early as the 1950s, it was observed that Ads were capable of killing HeLa cells [150,151], which led to recombinant attenuated strains of Ads being tested as oncolytic agents against various types of cancer. With the development of genetic engineering techniques, conditionally replicating adenoviruses (CRADs) have been constructed to target tumor cells, leading to an increase in pre-clinical studies of OVs targeting PCa at the start of the 21st century. Despite the tumorophilic nature of OVs, viral replication in normal cells is still inevitable. Scientists first focused on improving the targeting of lysing viruses to tumors and designed different CRADs based on the characteristics of PCa cells, placing essential viral genes under the control of PCa-specific promoters and increasing the enrichment of CRADs in PCa to enhance therapeutic efficiency. Meanwhile, other types of viruses with the ability to lyse tumors are gradually being discovered, and they are being tested in preclinical studies against PCa (Table 2).

#### 4.1.1. Preclinical Research to Develop New OV Viruses with Improved Replication and Cell Lysis Capabilities for PCa Treatment

##### Adenovirus

Recombinant OVs carrying tumor/tissue-specific promoters targeting PCa are a major strategy (Table 2). Since the tumor lytic effect was first discovered for disease-causing viruses, there has been a need to reduce the virulence of the virus and increase the specificity of tumor targeting. Therefore, at the time, in order to increase the effectiveness of prostate cancer therapy, scientists first considered increasing the specificity of lytic virus replication. In 1999, Henderson et al. constructed CV764 by inserting the prostate-specific antigen PSA enhancer and human kallikrein 2 (hK2) enhancer/promoter into the type 5 Ad genome. CV764 was observed to be significantly attenuated in primary human microvascular endothelial cells, exhibited a favorable safety profile, and was more targeted in PSA-producing LNCaP, human PCa cells compared to other tumor cells [152].

In the same year, the team constructed another oncolytic adenovirus, CV787 (also known as CG7870), which contained the rat probasin (PB) promoter-driven E1a gene and PSAe-driven E1B gene and a wild-type E3 area that suppresses the host immune system [153]. CV787 destroyed PSA-expressing cells 10,000 times more effectively than cells that did not express PSA, and a single tail vein injection of CG7870 eliminated LNCaP prostate cancer xenografts in athymic mice.

Tumor metastases in intermediate or advanced forms of PCa often localize in the bone. Prostate cancer cells from bone metastases express a non-collagenous bone matrix protein, human osteocalcin (hOC), which is expressed almost exclusively in bone and osteoblastic tumors. In 2001, a CRAD named Ad-hOC-E1a, whose viral replication-associated E1a gene was controlled by hOC expression, was constructed to target PCa cells in bone. This construct selectively replicates and exerts oncolytic effects in PCa cells in bone metastases and is effective against human PCa cell lines [154,155].

Another protein that is uniquely expressed in prostate epithelial cells and prostate cancer cells has been identified as a product of the alternative reading frame of the T cell receptor γ-chain (TARP). Essand et al. found that the regulatory sequence PPT, which contains the TARP promoter (TARPp), the prostate-specific membrane antigen PSMA enhancer (PSMAe), and PSAe, has high prostate-specific activity in the presence or absence of testosterone. They constructed Ad-PPT-E1a to specifically replicate and initiate tumorolytic effects in hormone-sensitive or hormone-insensitive PCa and showed remarkable results [156].

Ad.Δ55.HRE is a CRAD in which the E1B55 gene is missing and the expression of E1A is regulated by the hypoxia response element (HRE) expression system. Ad.Δ55.HRE expresses more E1A under hypoxia and exhibits significant antitumor activity in thymus-free mice carrying PC-3 PCa expressing hypoxia-inducible factor (HIF)-1α [157].

CN706 (also known as CG7060 or CV706) is a CRAD that drives the E1A gene by inserting PSAe into the type 5 Ad genome [158]. It has been reported that human PSA-producing LNCaP prostate cancer cells express high levels of E1A, which is absent in non-PSA-producing DU145 prostate cancer cells after infection with CN706. Notably, a single intratumoral injection with CN706 cured LNCaP xenografts and abolished PSA production in athymic mice [159].

Human telomerase reverse transcriptase (hTERT) is the catalytic component of the telomerase ribonucleoprotein complex, and the hTERT promoter is used to selectively drive transgenes in many hTERT-expressing human cancer cells, forming a constructing strategy of CRAD. In vitro studies in various cell lines have demonstrated that the replication of Ad-hTERTp-E1A, a CRAD that E1A driven by the hTERT promoter, is primarily restricted to telomerase-positive tumor cells, whereas no oncolytic activity is observed in normal primary fibroblasts and epithelial cells. Intratumoral use of the virus in thymus-free mice carrying human PCa xenografts results in significant inhibition of tumor growth and, in some cases, complete tumor regression [160]. In 2004, OAS403, which carries both the E2F-1 gene (a transcription factor that primarily upregulates genes associated with cell growth) and the hTERT promoter, was constructed and demonstrated that systemic administration in mice with LNCaP tumors resulted in complete tumor regression in over 80% of animals at tolerable doses [161]. OBP-301 (Telomelysin) is another CRAD that has been constructed according to this strategy, in which the hTERT promoter drives the expression of E1A and E1B genes linked with an internal ribosome entry site. Studies have determined that intratumoral injection of OBP-301 significantly inhibits LNCaP tumors in a subcutaneous model in nude mice [162].

DD3 is one of the most PCa-specific genes and has been used as a clinical bi-diagnostic marker. PTEN is frequently inactivated in primary PCa, which is critical to cancer progression. Insertion of PTEN into DD3-controlled OVs to form Ad.DD3.D55-PTEN, the mechanism of action is the induction of apoptosis. It was found that in mice with human PCa CL1, Ad.DD3.D55-PTEN administration resulted in a significantly smaller tumor size and almost complete inhibition of the tumor growth rate [163].

Moreover, carrying a prostate cancer-specific gene has been shown to enhance the safety and efficacy of CRAD and has demonstrated surprising results in animal experiments in addition. This shows that this strategy is effective, which has led scientists to continue exploring this research direction. Newer research has incorporated some genes with powerful toxicity into OVs.

OVs carrying suicide genes are less well studied for PCa but still have significant effects (Table 2). HSV thymidine kinase (HSV-tk), along with the drug ganciclovir (GCV), and cytidine deaminase (CD), along with the drug 5-fluorocytosine (5-FC), are two of the most widely used suicide gene systems [164]. In 1999, Freytag et al. targeted the CD/HSV-tk fusion gene to tumors with the virus by constructing Ad5-CD/TKrep and demonstrated the destruction of tumor cells in vitro while exhibiting minimal cytotoxicity to normal cells. Furthermore, the combination of CD/5-FC and HSV-tk/GCV autocidal gene systems has been shown to enhance the tumor cell-specific cytopathogenic effect of Ad5-CD/TKrep and to sensitize tumor cells to radiation therapy [165]. In 2006, Barton et al. constructed the second generation of OVs carrying a suicide gene named Ad5-yCD/mutTK(SR39)rep-ADP. Relative to first-generation Ad5-CD/TKrep, Ad5-yCD/mutTK(SR39)rep-ADP exhibited enhanced in vitro tumor cell killing and significantly greater tumor control in preclinical models of human cancer without increasing toxicity [166].

Transforming growth factor (TGF-β) plays a crucial role in the control of bone metastases from PCa [167,168,169,170]. PCa cells produce TGF-β, and high levels of circulating TGF-β1 and the TGF-β-dependent SMAD phosphorylation pathway in tumors are markers of poor prognosis [171,172,173,174]; therefore, targeting TGF-β is a promising strategy for treating PCa, as well as an attractive approach for bone metastasis. Ad.sTβRFc is an Ad5-based oncolytic adenovirus expressing soluble transforming growth factor beta receptor II-Fc fusion protein (sTGFβRIIFc), a protein that directly targets TGF-β and inhibits TGF-β signaling, and was first used in preclinical studies of breast cancer [175,176]. In a previous study, Ad.sTβRFc was injected via the tail vein into nude mice carrying PC-3-luc bone metastases, and tumor progression was significantly inhibited in the treated group compared to the buffered control group [177]. Owing to the hepatotoxic and systemically toxic of Ad5, Ad5/48 replaces Ad5 carrying sTGFβRIIFc to form mHAd.sTβRFc and, similar to Ad.sTβRFc, has a replication capacity in human PCa cells and expresses high levels of sTGβRIIFc. Compared to Ad.sTβRFc, mHAd.sTβRFc also inhibits tumor progression in nude mice carrying PC-3-luc PCa bone metastases and can be used at larger doses (4 × 10^11^ vs. 5 × 10^10^ viral particles/mouse), showing better safety [178].

Although Ads are the most studied OVs, there are more viruses with oncolytic capabilities, including those listed below, all of which have natural oncolytic effects and have been modified in previous studies.

##### HSV

G207 is an ICP6 gene-disabled HSV-1 [179]. G207 was first used in preclinical studies of malignant gliomas and was later shown to be effective not only in human PCa in vitro but also in inhibiting and eradicating more than 22% of tumors in thymus-free mice with human prostate cancer xenografts by intra-tumoral or intravenous viral injection [180,181]. Todo et al. constructed G47Δ by deleting the α47 gene in G207. As the α47 gene product is responsible for inhibiting transporters associated with antigen presentation (TAP), G47Δ can enhance MHC class I presentation [182]. Studies have demonstrated that G47Δ shows stronger anti-tumor activity in prostate cancer cells in vitro and in vivo compared to G207. More importantly, in thymus-free mice implanted with human hormone-sensitive prostate cancer, the combination of G47Δ and androgen ablation showed a summative effect, with greater inhibition of tumor growth than either treatment alone, as well as efficacy against CRPC [183].

Another HSV-1 virus variant, NV1020, was found to cause tumor lysis in all prostate cancer cell lines tested in vitro, as well as a reduction in serum PSA and tumor suppression following injection into thymus-free mice inoculated with PC-3 and C4-2 human prostate cancer [184]. The NV1023 virus, an HSV-1/HSV-2 oncolytic recombinant, was reported to inhibit primary tumor growth in TRAMP (transgenic adenocarcinoma of mouse prostate) mice after intravenous administration and lymph node metastasis [185].

Similar to CRAD, oncolytic HSV with human prostatic acid phosphatase xenolog (hPAP) was shown to enhance the killing effect on PAP-expressing tumors. Indeed, PAP has already been used for cancer vaccination in patients with PCa. Studies have demonstrated that treatment with hPAP-expressing HSV bPΔ6-hPAP not only significantly reduced tumor growth and increased survival in C57/BL6 mice bearing mouse TRAMP-C2 prostate tumors but also reduced the size of tumor nodules of mice bearing metastatic TRAMP-C2 lung tumors [186].

Transcriptional and translational dual-regulated (TTDR) viral essential gene expression can increase both viral lytic activity and tumor specificity. Lee et al. constructed two recombinant HSV-1, ARR(2)PB-ICP27 and ARR(2)PB-5’UTR-ICP27; ARR(2)PB is a prostate-specific promoter associated with transcription, 5’UTR is associated with translation, and ICP27 is a viral essential gene. In mice carrying human prostate tumors LNCaP, the tumor size was reduced by more than 85% by day 28 after a single-intravenous viral injection and no virus was detected in normal tissue as measured by real-time PCR analysis, confirming the safety and efficacy of this strategy [187].

##### Reovirus (Reolysin, Pelareorep)

Among all reoviruses, reovirus type 3 Dearing (T3D) shows mainly oncolytic activity. Compared to other OVs, reovirus has a clearer biomarker, Ras mutation and activation, due to the inhibition of double-stranded RNA activation of PKR in Ras-activated cells. In non-Ras-activated cells, the presence of viral transcripts leads to PKR autophosphorylation, which in turn inhibits viral protein synthesis and thus prevents viral replication. Ras-activated cells inhibit the autophosphorylation of PKR, maintaining it in an inactive state and allowing viral translation, replication and tumor lysis to occur, which leads to virus-mediated cancer cell death. Mutations involving Ras occur in approximately 30% of all human cancers, suggesting that the reovirus has the potential for widespread use [188,189,190]. In 1998, Coffey et al. reported that a single intratumoral injection of reovirus resulted in tumor regression in 65–80% of mice [191], and since then, the reovirus preparation Reolysin has been used in preclinical and clinical studies [192]. 

Preclinical studies of reovirus against PCa were conducted later than those of Ads. In 2010, Morris et al. found that reovirus caused the death of human prostate cancer cells PC-3, LNCaP and Du-145 by apoptosis, and tumor reduction was observed after a single intratumoral injection of the virus in nude mice bearing human prostate cancer xenografts [193]. Wild-type reovirus binds to cancer cells through the interaction of the viral stinger protein Sigma-1 with sialic acid and junctional adhesion molecule A (JAM-A). JAM-A expression is usually reduced in solid cancers, which is associated with poorer survival and a poorer prognosis. Reovirus jin-3 is a Reovirus mutant that infects cancer cells independently of JAM-A and has been validated in the human PCa cell lines PC-3M-Pro4luc2, DU145 and 22Rv1, representing a more promising therapeutic modality [194].

##### Newcastle Disease Virus (NDV) 

NDV is an avian paramyxovirus, of which 73-T is the most well-characterized oncolytic strain. Phuangsab et al. found that NDV inhibited the growth of several solid tumors, including human PCa (PC-3), by a single intratumoral injection in nude mice [195]. NDV infection depends on the cleavage of the fusion (F) protein. Shobana et al. constructed a recombinant NDV that could only be infected in PCa by altering the cleavage site of the F protein so that it could be cleaved by PSA and showed good lysis and specificity [196]. 

##### Vaccinia Virus (VV)

The vaccinia virus, widely known for its use as a vaccine to eradicate smallpox, was later found to have a killing effect on tumor cells as well. The safety and immunogenicity of PSA-expressing recombinant vaccinia virus in rhesus monkeys were reported as early as 1995 [197]. A recombinant vaccinia virus, GLV-1h68 2010, was demonstrated to have an oncolytic effect on PC-3 and Du-145. and a single intravenous injection of GLV-1h68 resulted in a significant reduction in PC-3 and DU-145 tumor xenograft models. Additionally, GLV-1h68 infection resulted in strong inflammatory and oncolytic effects, leading to a dramatic reduction in regional lymph nodes metastasized by PC-3 [198].

##### Measles Virus (MV)

Live attenuated measles virus (MV) vaccine strains have also been reported to possess oncolytic activity. Liu et al. constructed a recombinant MV by inserting a single-chain antibody (scFv) specifically against the extracellular structural domain of PSMA as a C-terminal extension of the MV attachment protein. They showed that recombinant MV infected and replicated only in PSMA-positive PCa and induced tumor regression in nude mice bearing LNCaP and PC-3 tumor xenografts [199]. Another strain of MV derived from the Edmonston (MV-Edm) vaccine strain showed considerable oncolytic activity against various solid tumors and hematological malignancies. Msaouel et al. demonstrated the susceptibility of PC-3, DU-145, and LNCaP to MV infection using MV-Edm expressing green fluorescent protein (GFP), and indicated that intratumoral administration inhibited tumor growth in nude mice carrying subcutaneous PC-3 xenografts [200].

##### Other Viruses

In addition to the abovementioned viruses, several other viruses are known to possess oncolytic effects. Three low pathogenic enteroviruses, the bioselective variants of coxsackievirus A21 (CVA21), coxsackievirus A21 (CVA21-DAFv) and echovirus 1 (EV1), have been reported to induce a reduction in xenograft tumor load in vivo following systemic delivery [201]. Respiratory syncytial virus (RSV) was found to have oncolytic activity against PCa (PC-3) both in vivo and in vitro, and intratumoral and intraperitoneal injections of RSV in nude mice resulted in a significant regression of PC-3 prostate cancer in vivo [202]. Inactivated Sendai virus (Hemagglutinating virus of Japan, HVJ-E) was also found to have oncolytic activity, with a significant in vitro killing effect on PC-3 and Du145. Direct injection of HVJ-E into the PC3 tumor cells of SCID mice reduced tumor volume and the disappearance of tumors in 85% of the mice [203]. Vesicular stomatitis virus encoding the SV5-F protein is replication-restricted and has been observed to induce apoptosis in PCa, showing oncolytic activity [204]. VSV-GP, a vesicular stomatitis virus carrying the glycoprotein of lymphocytic choriomeningitis virus, was also observed to effectively infect six PCa cell lines, and tumor remission was confirmed after intratumoral injection of the virus in a subcutaneous Du-145 tumor model [205]. Non-toxic Semliki Forest virus SFV-VA7, an oncolytic alphavirus, was reported to have a killing effect in the human PCa cell lines VCaP, LNCaP and 22Rv1, but not in the non-malignant prostate epithelial cell line RWPE-1. A single intraperitoneal injection of SFV-VA7 eradicated all subcutaneous and in situ LNCaP tumors [206].

Overall, the strategy of carrying specific genes has yielded promising modifications. The vast majority of recombinant OVs have shown considerable specificity and promising effectiveness in animal studies, and the use of highly virulent genes may further enhance the direct lytic effect of OVs. This strategy first ensures the safety of OVs, as well as their efficacy, and then proceeds to further studies on this basis, which agrees with the principles of phase I/II. Recombinant OVs are not only effective for PCa in situ but also for bone metastases; thus, this strategy is not only effective for early PCa but may also be able to address advanced mPCa. This strategy can greatly enhance the effectiveness of wild lysoviruses, domesticating the “uncontrollable” viruses and is still in use today. The development of this strategy has also led to a constant updating of the relevant technologies, which has helped expand the fields of cytology and virology. Unfortunately, at that time, scientists had not yet realized the power of the anti-tumor immunity of OVs, which is key to tumor eradication.

#### 4.1.2. Preclinical Research to Enhance the Immunostimulatory Effect of OVs for 

##### Treating PCa

As OVs continue to be studied, their role of OVs in inducing local and systemic immune killing of tumor cells is gradually being recognized as a more potent anti-tumor mode than direct tumor lysis [207]. Analysis of The Cancer Genome Atlas (TCGA) data also revealed that the low somatic TMB (Tumor Mutational Burden) of PCa reduces neoantigen expression compared to many other cancers [208], which is consistent with the immune microenvironment of “cold” tumors. Therefore, over the past decade, the design of recombinant OVs by adding different immunostimulatory molecules or other means to amplify their immune-inducing effects has become a novel means of enhancing the oncolytic properties of OVs.

IL-2 and other members of the IL family play a significant role in the OV-mediated killing of PCa cells through immune stimulation. Melanoma differentiation-associated gene-7/interleukin-24 (mda-7/IL-24) is a member of the IL-10 family of cytokines and has been confirmed to have anti-angiogenic, radiosensitizing, immunostimulatory and anti-tumor bystander effects [209,210,211], as well as inducing growth inhibition and apoptosis in various human cancers, including PCa, without any harmful effects on normal cells [212,213,214,215]. Studies have demonstrated that OVs carrying IL-24-expressing genes induce apoptosis by activating caspase-8 through the TLR3 pathway, thereby killing PCa cells more effectively in vivo and in vitro, inducing immunity, and inhibiting angiogenesis in vivo [216,217]. IL-12 can induce IFN-γ expression, promote immune cell survival and activation, enhance productive antigen presentation, and inhibit tumor angiogenesis. Preclinical studies have demonstrated that recombinant OV-IL-12 or OVs expressing IL-12 themselves enhance the anti-tumor efficacy of the virus against PCa through the above mechanism. Varghese et al. compared the efficacy of five recombinant HSVs (G207, G47Δ, NV1023, NV1034, and NV1042) against PCa, with NV1034 expressing GM-CSF and NV1042 expressing IL-12. The results showed that by days 6 and 10 post-treatment, only NV1042-inoculated tumors contained CD8^+^ and CD4^+^ cells, demonstrating superior anti-tumor immunity, while NV1042-treated tumors showed a significant reduction in CD31 staining, implying an effective anti-angiogenic effect of IL-12 [218]. Further studies have shown that IL-12-expressing OVs, along with other anti-tumor strategies, can enhance the anti-tumor effect on PCa [219,220,221,222,223].

Not only the IL family but also other substances may increase the recruitment of CD8^+^ T cells and thus enhance the anti-tumor immunity of OVs. CD40L stimulates DCs in PCa to activate CTL against the tumor, thereby improving the immunosuppressive phenotype in PCa [224,225,226]. A stroma-targeted bispecific T-cell splice agent (BiTE) that binds both CD3ε on T cells and fibroblast-activating protein on cancer-associated fibroblasts (CAF) upon expression via lysis group B Ad-enadenotucirev activates T cells in tumors and causes CAF death. The investigators observed that this recombinant virus induced IFN-γ production in PCa and thus promoted CTL activation, effectively producing pro-inflammatory effects and reversing the immunosuppression involved in CAF, thereby enhancing the anti-tumor capacity of OVs [227,228]. A PCa-specific oncolytic adenovirus (Ad-PL-PPT-E1A) with a prostate-specific antigen and CD40 ligand fusion gene has also been shown to induce apoptosis and to lead to specific lytic toxicity in PCa cells. Expression of CD80, CD83, CD86 and mRNA levels of IL-6, IL-12, IL-23, and tumor necrosis factor-α have been found to be significantly upregulated in Ad-PL-PPT-E1A-infected LNCaP cell lysates, demonstrating an effective immunostimulatory effect. Ad-PL-PPT-E1A treatment significantly increased survival and inhibited tumor growth in a human PCa (PC-3M) cell xenograft mouse model [229].

In addition to the Ads, other OVs have also been observed to have a notable immunostimulatory capacity to effectively produce anti-tumor effects on PCa in recent preclinical studies and enhancing their anti-tumor immunity has become a new strategy. Recombinant poliovirus PVSRIPO is a live vaccine that has been observed to have strong cytotoxic and innate inflammatory effects. PVSRIPO proliferated in Du-145 cells in vitro and was effective in lysis. Indeed, a single intratumoral injection of PVSRIPO was found to be effective in shrinking subcutaneous Du-145 tumors and improving survival rates. PVSRIPO-treated tumors were found to have significantly increased expression of pro-inflammatory chemokines and cytokines in 24 h, as well as increased immune cell infiltration within the tumor, suggesting that activation of anti-tumor immunity by PVSRIPO is a key factor in its efficacy [230]. NDV induces ICD and leads to the release of DAMP, which triggers an anti-tumor immune response. Xueke Wang et al. found that the combination of an inhibitory signal transducer and activator of transcription 3 (STAT3) and NDV enhanced ICD in NDV-induced PCa, thereby enhancing anti-tumor immunity and improving NDV-based anti-PCa effects [231].

**Table 2 ijms-23-12647-t002:** Pre-clinical research of OVs on PCa.

Name	Change of Construct (Compared to the Original Virus)	Time	Reference
**Original virus: Ads**			
CV764 (CG7060)	+PSAe, +hK2	1999	[152]
CV787 (CG7870)	+PB, +PSAe	1999	[153]
Ad5-CD/TKrep	+cytidine deaminase (CD), +HSV thymidine kinase (HSV-tk)	1999	[165]
Ad-OC-E1a	+OC	2001	[154]
Ad.Δ55.HRE	+hypoxia response element (HRE)	2004	[157]
Ad-hTERTp-E1A	+hTERTp	2004	[160]
OAS403	+hTERTp, +E2F-1	2004	[161]
Ad5-yCD/mutTK(SR39)rep-ADP	+yeast cytosine deaminase (yCD)/mutant (SR39), +adenovirus death protein (ADP)	2006	[166]
CN706/Ad5-PSAe	+PSAe	2007	[158]
OAS403&OBP-301	+hTERTp	2004&2008	[161,162]
Ad.DD3-E1A-IL-24	+IL-24	2010	[216]
Ad.DD3.D55-PTEN	+PTEN	2012	[163]
Ad.sTβRFc	+sTGFβRIIFc	2012	[177]
Ad-PL-PPT-E1A	+PPT, +CD40L	2014	[229]
Ad5/48	+sTGFβRIIFc	2014	[178]
Enadenotucirev-BiTE	+BiTE	2018	[227]
**Original virus: HSV**			
G207	−ICP6	1995	[179]
G47Δ	−α47	2001	[182]
NV1020	−15-kb region at the UL/S junction	2002	[184]
NV1034	+GM-CSF	2006	
NV1042	+IL-12	2006	
NV1023	+HSV-1/HSV-2 recombinant	2007	[185]
bPΔ6-hPAP	+hPAP	2010	[186]
ARR(2)PB-ICP27	+ARR(2)PB	2010	[187]
ARR(2)PB-5’UTR-ICP27	+ARR(2)PB, +5’UTRs	2010	[187]
**Original virus: Reovirus**			
Reovirus	—	2010	[193]
Reovirus jin-3	+jin-3	2022	[194]
**Original virus: NDV**			
NDV	—	2001	[195]
NDV	—	2013	[196]
**Original virus: VV**			
GLV-1h68	+Ruc-GFP, +β-galactosidase, +β-glucuronidase	2010	[198]
**Original virus: MV**			
MV-Edm	+J591 scFv	2009	[199]
MV-Edm	+GFP	2009	[200]
**Original virus: CVA21**			
CVA21	—	2008	[201]
CVA21-DAFv	+DAFv	2008	[201]
**Original virus: Echovirus 1**			
EV1	—	2008	[201]
**Original virus: RSV**			
RSV	—	2009	[202]
**Original virus: HVJ-E**			
HVJ-E	—	2009	[203]
**Original virus: VSV**			
VSV-SV5-F	+SV5-F	2010	[204]
VSV-GP	+GP	2018	[205]
**Original virus: Poliovirus**			
PVSRIPO	—	2016	[230]

+: Adding genes in original virus to get recombinant virus; −: Cutting genes in original virus to get recombinant virus; Ads: adenovirus; HSV: herpes simplex virus; NDV: newcastle disease virus; VV: vaccinia virus; MV: measles virus; CV: Coxsackievirus; RSV: respiratory syncytial virus; HVJ-E: hemagglutinating virus of Japan; VSV: vesicular stomatitis virus.

### 4.2. Clinical Trial of OVs on PCa

#### 4.2.1. Clinical Trial of Ads

Clinical studies using OVs for PCa date back to 2001, when DeWeese et al. conducted a phase I clinical study using CV706 in 20 patients with PCa recurring after radical radiotherapy at increasing viral doses from 1 × 10^11^ to 1 × 10^13^ (Table 3). This study reported the safety of CV706 in a phase I trial and validated its efficacy of CV706 against PCa in a phase I trial by reporting a greater than or equal to 50% reduction in PSA in patients treated with the highest viral dose [232].

The oncolytic adenovirus CV787 has also been used in clinical trials (Table 3). In the phase I trial, 23 mCRPC patients with mCRPC received a single intravenous dose of CV787 (highest dose 6 × 10^12^) and five patients showed a decrease in PSA of between 25% and 49%, with no patient showing a decrease in PSA of more than 50%. Notably, cytokines, such as IL-1, IL-6, IL-10, and TNF-α were observed to increase to varying degrees in the peripheral blood of the subject patients, which may indicate that CV787 activated immunity in the subject patients [233].

Ad-mediated tumor killing by suicide gene expression in tumors has been demonstrated in preclinical studies. Ad5-CD/TKrep targets the cytosine deaminase/herpes simplex virus-1 thymidine kinase fusion gene to tumors along with the virus, which not only sensitizes malignant cells to specific drugs but also makes them sensitive to radiation [165]. The phase I study was initiated in 2002 and included four cohorts of 16 PCa patients with local recurrence after radical radiotherapy, 10^12^ doses of virus were injected into the patients’ prostates by intratumoral administration, and the patients were treated with 5-fluorocytosine and ganciclovir 2 days later (Table 3). The results showed that 10 patients had varying degrees of reduction in serum PSA, while two patients were adenocarcinoma negative at 1 year follow up [234]. A new phase I clinical study in 2003 involved the Ad5-CD/TKrep concomitant 5-fluorocytosine and valganciclovir premedication combined with conventional dose 3D conformal radiotherapy in 15 newly diagnosed patients with moderate to high PCa. The results showed a significant decrease in prostate-specific antigen (PSA) in all patients, with a median follow-up time of 9 months [235]. In 2007, the team used a second generation virus Ad5-yCD/mutTKSR39rep-ADP2007 equipped with a modified yeast cytosine deaminase (yCD)/mutant SR39 herpes simplex virus thymidine kinase (HSV-1 TKSR39) fusion gene (yCD/mutTKSR39) along with intensity-modulated radiotherapy (IMRT) on a phase I trial (viral dose 10^11^–10^12^) in patients with intermediate to high risk PCa (Table 3). Seven of the remaining eight were negative at the last prostate biopsy, except for one patient who died midway through the trial due to complications from abdominal hernia surgery [236]. The team confirmed the efficacy of locally administered CRAD on PCa through clinical studies and follow-up, resulting in a survival benefit for patients [234,235,236,237]. Unfortunately, the CRAD was constructed to act more as a vector in this strategy; the suicide gene shows greater destruction of PCa than the OVs and does not fully reflect the direct lytic and immune-activating properties of OVs. The team used the precision replication properties of CRAD to deliver the suicide gene into the tumor via a virus, although other viruses can, not necessarily OVs, achieve the same effects.

#### 4.2.2. Clinical Trial of HSV-1

G47Δ is the only HSV-1 species virus for which clinical trials treating PCa have been reported in the literature. The phase I study of G47Δ in patients with CRPC started in May 2013 and was completed in 2016 (www.umin.ac.jp, UMIN000010463), representing the first use of an oncolytic HSV-1 in clinical trials for treating PCa (Table 3). In this single-arm study, G47Δ (3 × 10^8^ pfu) was injected into the prostate by a transrectal, ultrasound-guided, trans-perineal technique. The treatment was well tolerated by patients and no serious adverse events attributable to G47Δ have been observed so far [238].

#### 4.2.3. Clinical Trial of Reovirus

Oncolytic wild-type reovirus pelareorep (REOLYSIN) has been evaluated in a phase I trial in patients with advanced solid tumors, including PCa (Table 3). A total of 33 patients, including five with prostate cancer, were enrolled in the trial, all of whom had already received conventional treatment. The viral dose injected was 1 × 10^8^ to 3 × 10^8^ TCID_50_ by vein. The results demonstrated the safety of pelareorep at 3 × 10^8^ TCID_50_ doses, as well as the safety of intravenous and repeat dosing, and the reduction in PSA levels (from 100 to 50 ng/mL) in a patient with PCa indicative of potential efficacy [239]. A phase II trial of pelareorep was conducted in patients with stage T2 (organ-confined) PCa who received a single intraprostatic injection of pelareorep 3 weeks before prostatectomy. In this study, four of the six tumors showed evidence of tumor cell apoptosis, while one patient had decreased PSA levels. No toxicity was observed, and immune cell infiltration was limited to tumors [240].

#### 4.2.4. Clinical Trial of Vaccinia Virus

The recombinant vaccinia virus expressing prostate-specific antigens is used in multiple clinical trials for the therapy of PCa (Table 3). A phase I trial of rV-PSA (vaccinia virus carrying PSA gene) was evaluated in 33 men with elevated PSA levels after radical prostatectomy, radiation therapy or the development of metastatic disease. Patients received three doses of rV-PSA (2.65 × 10^6^ to 2.65 × 10^8^) continuously monthly and the 10 patients who received the highest dose also received 250 ug/m^2^ granulocyte-macrophage colony-stimulating factor (GM-CSF) as an immunostimulatory adjuvant. The PSA levels in all 33 men treated with rV-PSA, with or without GM-CSF, remained stable for a period of time and did not progress [241]. 

In another phase I trial of rV-PSA in patients with advanced PCa (Table 3), 42 patients received 2.65 × 10^5^ to 2.65 × 10^8^ doses of rV-PSA by subcutaneous injection. The results showed no objective tumor response, the mean time to disease progression was 106 days (range 28–310 days), 6 of 42 patients had stable disease, 30 had progressive disease and 6 were not assessed [242]. 

A phase II clinical trial evaluating three vaccinia viruses, including rV-PSA, in advanced prostate cancer enrolled 64 patients (Table 3). The results revealed no serious adverse effects, with 2.34 × 10^8^ PFU of rV-PSA by intradermal injection or 1.5 × 10^9^ by intramuscular injection of the vaccine at 6-week intervals. Moreover, 45.3% of the men remained free of PSA progression at 19.1 months, while 50 men showed no clinical progression [243]. These clinical trials provide preliminary evidence of the safety and efficacy of rV-PSA [242,243,244,245,246,247]. At this time, recombinant vaccinia virus was seen more as a cancer vaccine (such as inactivated viruses or viral antigens) than as an oncolytic agent; indeed, its oncolytic effect was ignored in favor of anti-tumor immunity and continued to be investigated as a novel agent for treating mCRPC.

#### 4.2.5. Clinical Trial of HVJ-E

Inactivated Sendai virus HVJ-E was investigated in an open-label, phase I/II dose-escalation study in patients with docetaxel-resistant CRPC (Table 3). Seven CRPC patients with CRPC were included in the study; four patients were assigned to the low-dose group, and three patients were assigned to the high-dose group. HVJ-E was injected directly into the prostate by transrectal ultrasound guidance with a PEIT (percutaneous ethanol injection therapy) needle on day 1, and then again on days 5, 8 and 12 of the 28-day treatment cycle of subcutaneous injections. One patient in the low-dose group withdrew from the treatment halfway through. Assessment of the PSA response to treatment in six patients showed that complete remission was found in one patient in the low-dose group, while PSA levels in the other five patients were not reduced after two cycles of HVJ-E treatment. On imaging, according to RECIST criteria, three patients in the low-dose group had stable disease, one patient in the high-dose group had stable disease and two patients had disease progression. No serious adverse events were reported in the clinical trials, indicating that intratumoral and subcutaneous injections of HVJ-E are well tolerated and feasible. Unfortunately, no changes in serum IFN-α, IFN-β, IFN-γ, and IL-6 levels and NK cells were observed during the treatment period [248]. 

Another phase I clinical trial evaluated HVJ-E (GEN0101) in patients with mCRPC (Table 3). Nine patients with mCRPC were divided into two groups of high (six patients) and low (three patients) doses and were administered intratumorally and subcutaneously in the groin. Although the serum PSA levels did not decrease in either group, the PSA growth rate was more significantly inhibited in the high-dose group than in the low-dose group. Moreover, two patients in the low-dose group and five patients in the high-dose group showed an increase in NK cell activity of more than 10%, but there were no significant changes in the serum IFN-γ and IL-6 levels [249].

With the review of reported phase I and II clinical studies of OVs for patients with PCa, most adverse events experienced by patients were Grade 1 or 2, with very few Grade 3 adverse events and no associated Grade 4 adverse events or patient deaths due to the drug. Injection site reactions (Grade 1) were the most frequent adverse events. The results of individual clinical trials show that OVs activate immune anti-tumors in humans. Unfortunately, the vast majority of clinical trials of OVs for PCa have been administered by local injection into the tumor, and the safety and efficacy of intravenous injection have not yet been explored. Intravenous OVs represent a less invasive mode of administration than local injections and may effectively target free tumor cells in the peripheral circulatory system and act via the circulatory system on both primary and metastatic sites, which has the potential to lead to more intense adverse events.

**Table 3 ijms-23-12647-t003:** Clinical Trials of OVs on PCa.

Name	Time	Phase	Tumor Types	Number of Patients	Concentration	Delivery Way	Reference
**Adenovirus**							
CV706 (CG7060)	2001	I	Local recurrence of PCa after radical radiotherapy	20	10^11^–10^13^	Intratumoral	[232]
CV787 (CG7870)	2006	I	mCRPC	23	1 × 10^10^–6 × 10^12^	Intravenous	[233]
Ad5-CD/TKrep	2002	I	Local recurrence of PCa after radical radiotherapy	16	10^12^	Intratumoral	[234]
Ad6-CD/TKrep	2003	I	Newly diagnosed med-high risk PCa	15	10^12^	Intratumoral	[235]
Ad5-yCD/mutTKSR39rep-ADP	2007	I	Newly diagnosed PCa	9	10^11^–10^12^	Intratumoral	[236]
**HSV-1**							
G47Δ	2013–2016	I	CRPC	9	3 × 10^8^ pfu	Intratumoral	[238]
**Reovrius**							
Pelareorep (REOLYSIN)	2008	I	PCa have been treated	5	3 × 10^8^ TCID_50_	Intravenous	[239]
Pelareorep (REOLYSIN)	2006	II	T2 PCa	6	-	Intratumoral	[240]
**Vaccinia viru**							
rV-PSA	2000	I	PCa with elevated PSA after radical surgery or metastasis	33	2.65 × 10^6^–2.65 × 10^8^	Intratumoral	[241]
rV-PSA	2002	I	Advanced mPCa	42	2.65 × 10^5^–2.65 × 10^8^	Subcutaneous	[242]
rV-PSA	2004	II	Advanced PCa	64	2.34 × 10^8^/1.5 × 10^9^	Subcutaneous/intramuscular	[243]
**HVJ-E**							
HVJ-E (GEN0101)	2017	I/II	CRPC	7	3000/10,000 mNAU	Intratumoral/Subcutaneous	[248]
HVJ-E (GEN0101)	2020	I	mCRPC	9	3000/6000 mNAU	Intratumoral/Subcutaneous	[249]

## 5. Summary

Over the past decade, research on OVs for treating tumors has continued to progress, with their anti-tumor mechanisms being elucidated and refined and some satisfactory results obtained from clinical studies. For urological tumors, the treatment of advanced PCa is a challenge for professionals in the field, with first-rate morbidity, hormone resistance and a short patient overall survival making the current ADT-based combination of therapies seem overwhelming, with a negative impact on treatment. To this end, immunotherapy for PCa has evolved, with PD-1/PD-L1, and Sipuleucel-T, a DC-based autologous cellular immunotherapy, both approved by the FDA and used for treating patients with PCa [249,250,251,252,253,254,255,256,257,258].

As anti-tumor agents with biological properties, the most important feature of OVs is their powerful immune activation capabilities, which increase immune infiltration in the tumor microenvironment. The mechanism and ability to potentiate anti-tumor immunotherapy has been elucidated in the main text, and this “immune lighter” property gives OVs great potential for combining with other immunotherapies to enhance the efficacy of immunotherapy in treating PCa as a “cold tumor.” OVs have been studied as viral vectors targeting PCa for over a decade and have achieved promising results in preclinical studies and clinical trials. As anti-tumor immune mechanisms are now the main part of research in OVs and the mechanisms of efficacy overlap with other immunological agents have been revealed, OVs may later occupy a crucial place in the immunotherapy of PCa.

However, OVs also have shortcomings compared to conventional drugs. Indeed, some tumor patients, especially those with advanced tumors, are cachectic and thus may lack adequate nutrition and an intact immune system to fight off viral invasion. OVs have a viral nature and may cause a violent antiviral response in patients, further damaging their health. At this point, OVs are no longer healing drugs but deadly poisons. Therefore, it is crucial to ensure that weakened patients are tolerant of OVs.

OV-related agents are no longer only at the experimental stage, and the Republic of Latvia approved Rigvir (ECHO-7 virus) for use in clinical melanoma, colorectal cancer, pancreatic cancer and bladder cancer in 2004. Moreover, Ankeri (recombinant human Ad5) was approved for use in clinically advanced, recurrent head and neck tumors by the NMPA of the State Drug Administration of China in 2005. Imlygic (T-VEC) was approved by the US FDA in 2015 for treating clinically advanced melanoma. Furthermore, the Japanese Ministry of Health, Labor and Welfare (MHLW) approved Delytact (G47Δ) for use in patients with clinical malignant glioma on 1.11.2021. The marketing and clinical use of OV formulations demonstrate their safety, reliability and potential. Although the formulation of OV agents for PCa is still in the developmental stage, prospects for the use of appropriate therapy are promising.

## Figures and Tables

**Figure 1 ijms-23-12647-f001:**
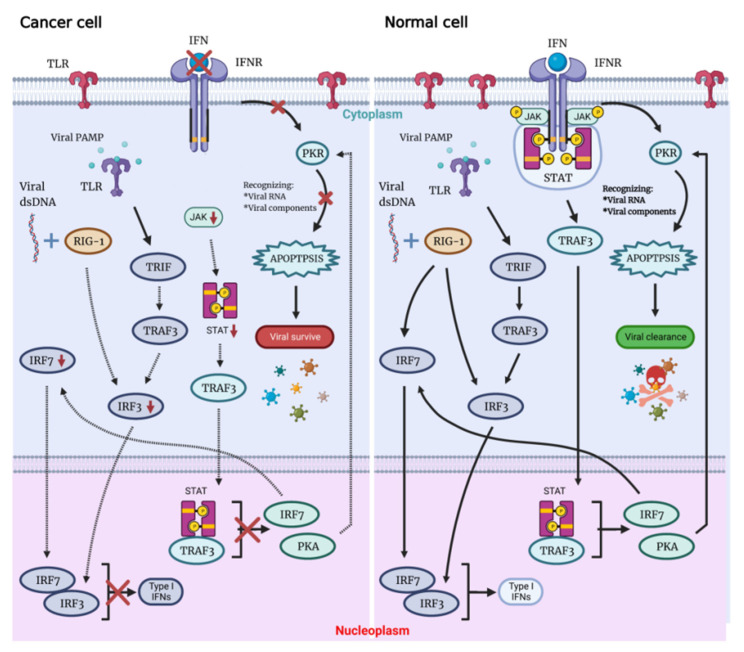
The OVs can specifically lyse tumor cells, sparing normal cells. Following viral infection, most normal cells activate the antiviral pathway, thereby controlling the infection. The antiviral mechanism can be triggered by activating the Toll-like receptor and RIG-I-like receptor signaling by viral PAMPs, which might include viral nucleic acids. Once the virus is detected, a signaling cascade of several IFN components JAK, STAT and interferon regulatory factor 9 (IRF9) leads to a programmed transcription pathway that limits the spread of the virus and can target infected cells for apoptosis or necrosis. Local IFN production induced by the innate immune response to viral infection may also contribute to antiviral activity via IFNR. Type I IFN signals through the JAK-STAT signaling pathway lead to the upregulation of cell cycle regulators, such as PKR and IRF7. These regulators limit viral spread by binding to viral particles and triggering the type I IFN transcriptional pathway, promoting aborted apoptosis and cytokine production in infected cells.

**Figure 2 ijms-23-12647-f002:**
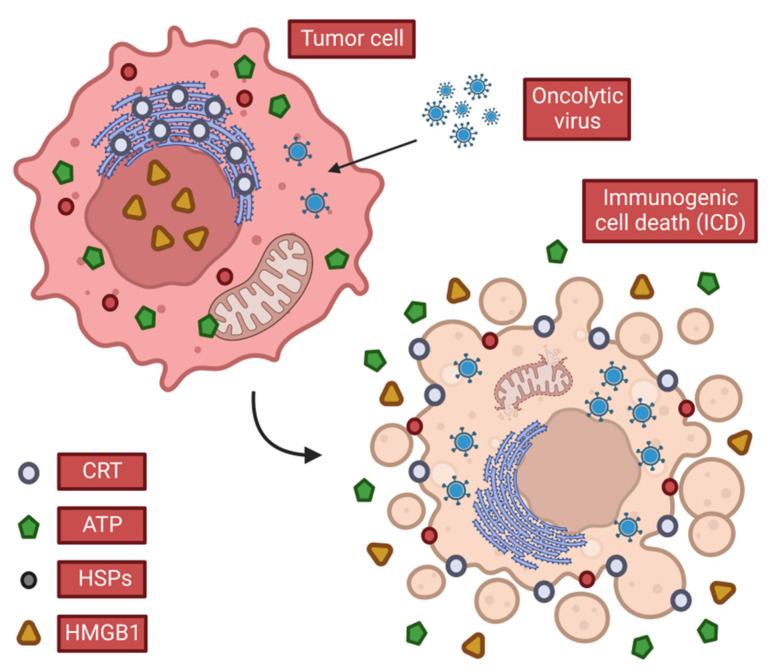
Characteristics of immunogenic cell death. When OVs cause ICDs in tumor cells, they release or expose DAMPs that stimulate anti-tumor immune responses. Tumor cells release ATP and HMGB1 in the extracellular space and expose CRT and heat shock protein (HSP), which promotes the continuation of anti-tumor immunity.

**Figure 3 ijms-23-12647-f003:**
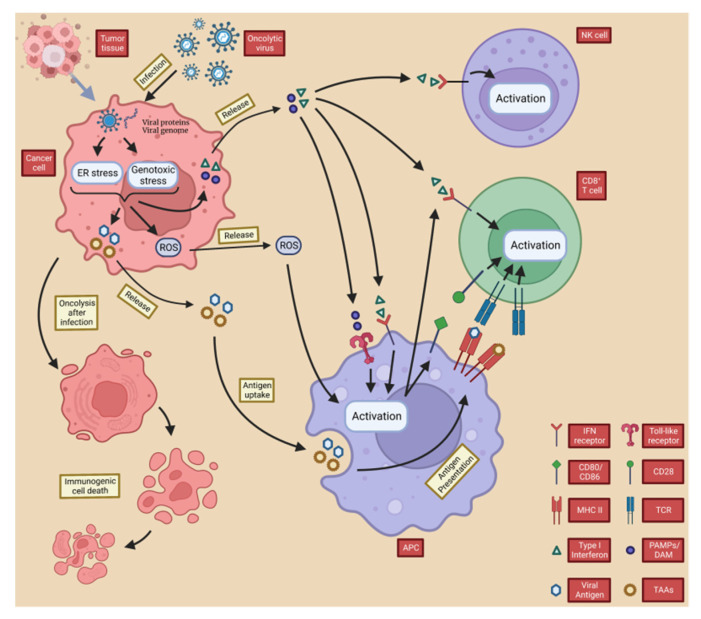
OVs induce local and systemic anti-tumor immunity. Following infection with the oncolytic virus, cancer cells initiate an antiviral response consisting of endoplasmic reticulum (ER) and genotoxic stresses, which results in the upregulation of reactive oxygen species (ROS) and the production of antiviral cytokines. ROS and cytokines, particularly type I interferon (IFN), are released from infected cancer cells and stimulate immune cells (antigen-presenting cells, CD8+ T cells, and NK cells). PAMP (made up of viral particles) and DAMP (made up of host cell proteins) stimulate the immune system by triggering activation receptors, such as TLRs. In the resulting immune-stimulatory environment, TAA and neoantigens are taken up and released by antigen-presenting cells.

**Table 1 ijms-23-12647-t001:** Classification of OVs.

Name	Baltimore Classification	Family	Capsid Shape
**DNA virus**			
Adenovirus	Group I: dsDNA	Adenoviridae	Icosahedral
Herpes simplex virus	Group I: dsDNA	Herpesviridae	Icosahedral
Vaccinia virus	Group I: dsDNA	Poxviridae	Complex
**RNA virus**			
Reovirus	Group III: dsRNA	Reoviridae	Icosahedral
Measles virus	Group V: ss(−) RNA	Paramyxoviridae	Icosahedral
Newcastle disease virus	Group V: ss(−) RNA	Paramyxoviridae	Helical
Vesicular stomatitis virus	Group V ss(−) RNA	Rhabdoviridae	Helical
Coxsackievirus	Group IV: ssRNA	Picornaviridae	Icosahedral
Poliovirus	Group IV: ss(+) RNA	Picornaviridae	Icosahedral

dsDNA: double-stranded DNA; ssDNA: single-stranded DNA; ss(+)RNA: positive single-stranded RNA; ss(−)RNA: negative single-stranded RNA.

## Data Availability

Not applicable.

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
