# Peer review of "Oncolyic Virotherapy for Prostate Cancer: Lighting a Fire in Winter"

_ijms, 2022, doi:10.3390/ijms232012647_

Round 1

Reviewer 1 Report

The manuscript submitted requires extensive language editing.

Author Response

Thanks for your Comments and Suggestions. I have editted the sentences of manuscript.

Reviewer 2 Report

This literature review is very extensive and detailed. This is both an advantage and a disadvantage. The text contains too much information, which is difficult to consume even by a person with an appropriate background. There is an overall good structure in the manuscript, but the logical structure within the subsections is sometimes very poorly expressed. Often it is simply a sequential description of publications on a topic without analysis. Perception can be greatly improved in the revision process. In the text, descriptions of publications and comparatively unimportant details replace the much more important analysis of these publications and the analysis of the trends of development of oncolytic virotherapy. It is desirable to remove from the text the details that are difficult to understand and difficult to remember. It is better to find a way to either get rid of these details or move them to additional tables. I would like to see more analysis in the review, revealing general trends in the successes as well as the obstacles that lie in the path of the development of oncolytic virotherapy for prostate cancer. It is not enough just to list the main findings from publications, it is necessary that the authors of the text analyze them at least to some degree. I have put my many edits and suggestions to improve the content, structure, and perception of the text in a pdf file. Here I summarize only general suggestions.

1)   I suggest structuring the information about viruses in such a way that viruses are grouped according to their characteristics and these characteristics are indicated. For example, it is possible to make the following sequence of narration which would follow this principle: viruses with the genome represented by DNA are listed first, and then viruses with the genome represented by RNA are listed. The latter type of viruses can be divided into groups according to weather their genomes are represented by plus or minus strands of RNA. When talking about each virus, it would be beneficial to mention the family it belongs and properties of family, the size of the genome and some other important biological characteristics of the virus.

2)   The text requires considerable improvement in English. It will benefit greatly if it is divided into paragraphs, following the golden rule of "one thought, one paragraph”. Numerous stylistic problems must be removed and semantic errors, which abound in the text, corrected. Numerous typos should also be corrected. Some sections are written better compared with other, for example 2.2, 3.1, 3.2.

3)   To improve comprehension, I suggest introducing more tables and figures (or both) that will summarize the information. Authors should think about which sections can be illustrated with figures or schemes. I suggest adding tables to the section 3 and 4, and the illustrations in a form of figures or schemes to following sections as candidates: 2.2, 2.3, 3.2, but authors might choose some of them.

4)   In addition, I suggest that the authors think about how to structure too long sections into additional subsections. For example, section 2.2. entitled OVs induce local and systemic anti-tumor immunity is too long. Perhaps some ideas can be expressed in subsections 2.2.1, 2.2.2 2.23, and so on. I also think that section 2.3 could become a subsection of section 2.2. Here I suggest that the authors think about it, because the content of the section 2.3 is also related to the title of section 2.2. Although the authors may have arguments against this proposal, and perhaps that they may be justified.

5)                  Try to add main results of a clinical trials to the tables, both in terms of side effects and effectiveness and remove extra details from the text. Find a way to add this information to existing tables or make additional tables. Highlight in the table the type of OV therapy with or without combination with another therapy. Add references to all tables.

Author Response

Thanks for your Comments and Suggestions. I have editted the sentences of manuscript as your requirements. And I also add one table to show the virus characteristics information, and three figures to explain the Mechanism of viral infection of tumor cells in manuscript. Also, I delete some detailed data and add lots of my own viewpoints in main text and improve the structure of the manuscipt. Finally, thanks your suggestions again, and I really learn a lot from it.

Reviewer 3 Report

 I appreciate and respect the author's efforts. The review was well written. It would be advantageous if the authors would include a few schematics to describe the mechanism of OV in mCRPCs.

Author Response

(The authors gave the same response as above.)

Reviewer 4 Report

Comments to authors

This manuscript provides a thorough overview of oncolytic virotherapies for prostate cancer. It describes the incidence and mortality of prostate cancer worldwide, the different stages of prostate cancer and their standard therapies, and the different viruses used for therapeutic treatment preclinically and clinically against prostate cancer. It also describes how DNA technology and innovations dramatically improved OV therapy for prostate cancer over the years, and the wide range of applications of combination therapies with chemotherapy, radiotherapy, gene therapy, and immunotherapy.

Overall, this is a very interesting and thorough review about oncolytic virotherapy for prostate cancer. An in-depth overview has been presented about the different viral constructs that has been developed over time and their mechanism(s) of action against cancer, as well as the incidence and mortality worldwide has been discussed. With the increasing interest in microbial treatment of cancer, this manuscript is particularly timely. However, some areas could be more extended as discussed below.

A section about safety issues of oncolytic virotherapy against cancer, as well as advantages and disadvantages of oncolytic virotherapy compared to other therapies would strengthen the manuscript. For instance, the number of injections, and the cytokine responses (that have been improved  over the years through genetic modification). A section with future prospects how to further improve oncolytic virotherapy in the near future would strengthen the manuscript as well.

Finally, an extra column in table 1 with the results of the different virotherapies, to determine which viruses are most promising against PCa will be very helpful.

Author Response

Thanks for your Comments and Suggestions. I have editted the sentences of manuscript, and add one table to show the virus information. Also three figures are showed the Mechanism of viral infection of tumor.

Round 2

Reviewer 2 Report

           The pdf file with corrections is attached. The authors of this text have made a substantial effort to remove the shortcomings of the review that were mentioned earlier. However, the review is still very difficult and sometimes impossible to comprehend.  The very first thing that catches your eye is the poor English. This is a very important and significant flaw. I mentioned this problem before. The authors of the review have read and analyzed a huge body of literature, but they do not manage to communicate the research they have read and analyze in a simple and comprehensible way. In doing so, the text of the review, especially its second part, is very complex, full of terms and meanings, as well as descriptions of small details of studies that were conducted long ago, 10 or even 20 years ago. Many of these small details - like names of genetic constructs - have not been used in the scientific literature for many years. When the reader is confronted with the names of the constructs that have not been used for a long time, it is immediately difficult to understand what authors of the manuscript are talking about. The explanation of the nature of the genetic construct is difficult to extract from very long and confusing sentences with poor grammar. There are sentences in a text that combine several separate sentences with different meanings.

             The first part of the review, which has fewer such problems, is a little easier to read and understand than the second. When faced with the complex sentences of the first part, it is possible to think and guess at the meaning of what the authors wanted to say. However, even in this case, many sentences a person must read many times and think for a long time, what exactly the authors of the manuscript wanted to say. As a reviewer, I know the subject matter and the research the manuscript is about. That's why it's easier for me to guess what the authors want to say. What will happen to the reader who is not an expert? He or she simply will not be able to wade through the obstacle of complex sentences with confusing meanings. In the first part of the manuscript, I have tried to highlight particularly difficult sentences and, in some cases, to correct them. However, my editing should be considered advisory, but not obligatory, because I may not have guessed correctly what the authors of the review wanted to say.

               I didn't fix or even comment the second part of the review, because even I can't wade through the jumble of complex sentences and terminology that haven't been used in a while.  However, the lack of my edits or comments does not mean that the text does not need to be corrected and it is digestible. On the contrary, this lack of edits indicates that the text is so complex and confusing that it needs the next level of effort and time to improve it, which I cannot provide.

               If the review is published as it is now, it is unlikely to have appreciative readers. The text is incredibly complex incomprehensible and confusing with lots of mistakes in English. The authors need to combine their efforts with professional specialists who are good at English. The work to improve English should be done together. Because if the text is simply left to the professionals to correct, the English may become correct, and the scientific meaning may be lost. 

             Combinations of the correctness of the expression of scientific ideas and with the correctness of language is highly important for scientific reviews. The whole point of writing such manuscripts is to get the communication right and clear.

               As a less but still important point there are no references in the text to Table 1, Figure 1, and Figure 2. The text of the review by itself, and the tables and figures by themselves, they are not mentioned in the text.

One more point: it is better to organize table 1 according to the principle of moving from more general categories to more specific ones, from left to right. For example, in this sequence: Baltimore class, family, capsid properties (I don't think it is important, but you can leave it) – and specific viral representative. In addition, it makes sense to combine members of the same family and group together. In other words, it is better to list members of the same viral family or group next to each other. For example, measles virus and Newcastle disease virus are paramyxoviruses and group V, and they belong to the same family.

Author Response

Dear Reviewer,

Following your suggestion, I have completed my revised manuscript.
I made linguistic revisions to the manuscript with the help of an American editor with extensive experience editing SCI articles in this academic field. Also, I simplified Part 2 and removed some irrelevant details.

Round 3

Reviewer 2 Report

Scientifically, the review is fine.